

# Research progress of aldehyde oxidases in plants

Jun Wu[1], Blair Moses Kamanga[1], Wenying Zhang[1], Yanhao Xu[2] and Le Xu[1]

[1] Engineering Research Centre of Ecology and Agricultural Use of Wetland, Ministry of Education, Yangtze University, Jingzhou, China
[2] Hubei Academy of Agricultural Science, Wuhan, China

## ABSTRACT

Plant aldehyde oxidases (AOs) are multi-functional enzymes, and they could oxidize abscisic aldehyde into ABA (abscisic acid) or indole acetaldehyde into IAA (indoleacetic acid) as the last step, respectively. AOs can be divided into four groups based on their biochemical and physiological functions. In this review, we summarized the recent studies about AOs in plants including the motif information, biochemical, and physiological functions. Besides their role in phytohormones biosynthesis and stress response, AOs could also involve in reactive oxygen species homeostasis, aldehyde detoxification and stress tolerance.

## INTRODUCTION

Abscisic acid (ABA) acts as a ubiquitous signal that regulates diverse processes and plant responses, including seed maturation/germination, dormancy, leaf senescence, stomatal aperture, apical dominance, and the regulation of phototropic and gravitropic behaviour, as well as adaptation to a variety of environmental stresses, such as drought and salinity (*Verslues & Zhu, 2005*; *Normanly, Slovin & Cohen, 1995*; *Merlot & Giraudat, 1997*; *Srivastava et al., 2017*).

ABA biosynthetic genes and enzymes have been identified in various plants species (*Taylor, Burbidge & Thompson, 2000*; *Milborrow, 2001*; *Seo & Koshiba, 2002*; *Nambara & Marion-Poll, 2005*). ABA synthesis starts from C40 epoxycarotenoids that derived from isopentenyl diphosphate in methylerythritol phosphate pathway. Zeaxanthin epoxidase (ZEP) is responsible for catalyzing the two-step epoxidation of all-trans-zeaxanthin to all-trans-violaxanthin (*Finkelstein, 2013*; *Marin et al., 1996*; *Audran et al., 1998*, *2001*; *Xiong, Schumaker & Jian-Kang, 2002*). The latter is then converted into the cis-isomers of neoxanthin and violaxanthin, which are cleaved by a 9-cis-epoxycarotenoid dioxygenase (NCED) to its direct precursor, xanthoxin (*Schwartz et al., 1997*; *Qin & Zeevaart, 1999*), and this oxidative cleavage process is a key rate-limiting step for ABA biosynthesis (*Estrada-Melo et al., 2015*). Conversion of xanthoxin to abscisic aldehyde is catalyzed by a short-chain dehydrogenase/reductase (SDR) (*Cheng et al., 2002*; *González-Guzmán et al., 2002*), and aldehyde oxidases (AOs; EC 1.2.3.1) oxidize abscisic aldehyde into ABA as

Corresponding author
Le Xu, 501140@yangtzeu.edu.cn

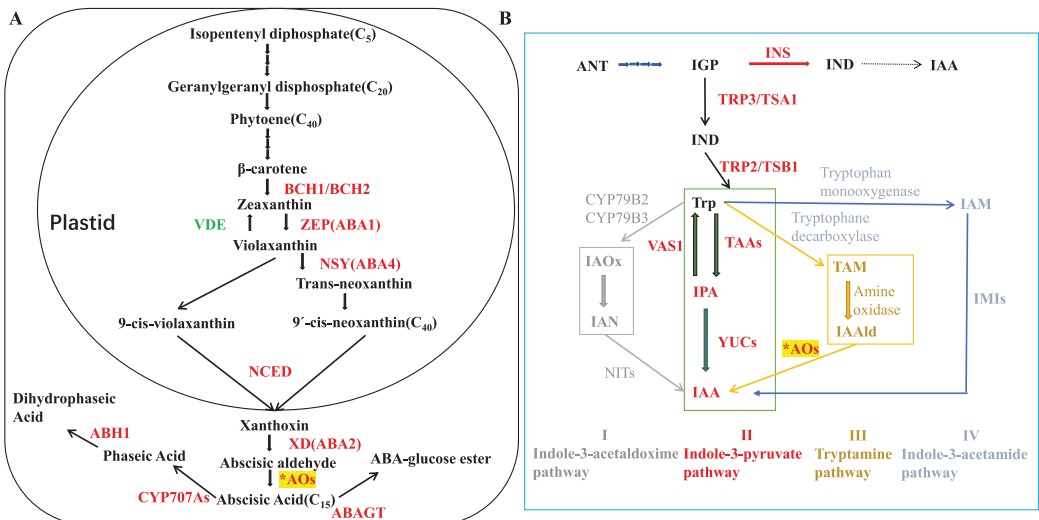

**Figure 1 ABA and IAA biosynthesis.** (A) ABA precursor is synthesized from the methylerythritol phosphate (MEP) pathway. Enzymes are shown in red colour. BCH1/BCH2: β-carotene hydroxylases; ZEP: Zeaxanthin epoxidase; NSY: Neoxanthin synthase; NCED: 9-cis-epoxycarotenoid dioxygenase; XD: Xanthoxin dehydrogenase; ABAO: Abscisic aldehyde oxidase; CYP707A: ABA 8′-hydroxylase; ABH1: Phaseic acid reductase 1; ABAGT: ABA glucosyltransferase; βG: β-glucosidase; VDE: violaxanthin de-epoxidase, AOs: aldehyde oxidases were indicated with an asterisk (*). Adapted from *Dejonghe, Okamoto & Cutler (2018)*; *Song et al. (2020b)*; *Finkelstein (2013)*. (B) ANT, anthranilate; IAA, indole-3-acetic acid; IAAld, indole-3-acetaldehyde; IAN, indole-3-acetonitrile; IGP, indole-3-glycerol phosphate; IND, indole; AAO, aldehydeoxidase; CYP79B2/3, cytochrome P450 monooxygenases2/3; IMI, amidase; INS, indole synthase; NIT, nitrilase; TAA, tryptophan aminotransferase; TRP3/TSA1, Trp synthase α-subunit; TRP2/TSB1, Trp synthase β-subunit; VAS1, pyridoxal phosphate-dependentaminotransferase1; YUC, YUCCA flavin-containing monooxygenase. Adapted from *Kasahara (2016)*; *Song et al. (2020a)*.

the final step (*Seo et al., 2000a*, *2000b*; *Barrero et al., 2006*; *Melhorn et al., 2008*; *Ando et al., 2006*). Pathway for ABA biosynthesis was summarized in Fig. 1A.

AOs are widespread cytosolic molybdo-iron-flavo enzymes that oxidize a variety of aldehydes to their corresponding carboxylic acids (*Koshiba et al., 1996*; *Ori et al., 1997*; *Seo et al., 1998*, *2000a*, *2000b*; *Garattini, Fratelli & Terao, 2008*; *Zdunek-Zastocka, 2008*). Except involved in ABA synthesis, AO enzyme could also oxidize indole acetaldehyde into indoleacetic acid (IAA, most important form of auxin). Though the tryptophan (Trp)-independent pathway has been proposed for IAA biosynthesis alternatively, Trp-dependent pathway is the main one including four proposed pathways for IAA biosynthesis in plants: (I) indole-3-acetaldoxime (IAOx) pathway, (II) the indole-3-pyruvic acid (IPA) pathway, (III) tryptamine (TAM) pathway, and (IV) the indole-3-acetamide (IAM) pathway. Based on evolutionarily conserved core mechanisms, it is thought that the pathway *via* IAM or IPA are the major route(s) to IAA in plants (*Mano & Nemoto, 2012*). AO is responsible for the final step of IPA pathway in auxin biosynthesis (Fig. 1B). Auxin is deeply implicated in most aspects of plant growth and development, and the detail of AO in IAA production will discuss in the part of aldehyde oxidase 1 later.

The nucleotide and amino acid of the AO multigene family shares high similarity with xanthine dehydrogenase (XDH), and AO is likely derived from XDH by gene duplication and neofunctionalization, though their function could be divergent. Both AO and XDH comprise FAD, Fe-S, and molybdenum cofactor (Moco) as prosthetic groups (*Koshiba et al., 1996*; *Hille, Nishinoab & Bittner, 2011*). However, different physiological electron acceptors distinguish AO and XDH enzymes: AO enzymes does not contain NAD-binding site, and AO can exclusively use molecular oxygen as a physiological electron acceptor and XDH use NAD$^+$ as the final electron acceptor (*Nishino & Nishino, 1989*; *Turner, Weiner & Tayior, 1995*).

Plants AOs catalyze the oxidation of a variety of different aromatic and aliphatic aldehydes (*Zdunek-Zastocka et al., 2004*). The conversion of abscisic aldehyde to ABA and indole acetaldehyde into indoleacetic acid by AOs, respectively, may constitute important regulatory element; however, its regulatory role is still being debated and we summarize the biochemical and physiological function of AO1, AO2, AO3, and AO4.

## SURVEY METHODOLOGY

Primary and secondary literature relevant to the topic of this paper was assessed using Web of Science (Clarivate Analytics). Key words such as "aldehyde oxidase," "abscisic acid," "indoleacetic acid," "plants" and "stress," were searched between 1 March and 31 October, 2021.

The information of stress induced genes, interaction proteins and function predictions were generated with the interaction viewer at bar.utoronto.ca/eplant by *Waese et al. (2017)*. The detail of experiment information can be found on this website. Take AO1 gene (AT5G20960) in *Arabidopsis* as an example, search the gene number, and choose Abiotic Stress II eFP, and we could find the total RNA and polysomal mRNA level was upregulated after hypoxia stress.

### Biochemical and physiological function of AO1

Plant AOs have been studied from several sources, including oat (*Avena sativa*) coleoptiles (*Rajagopal, 1971*), potato tubers (*Rothe, 1974*), cucumber (*Cucumis sativus*) seedlings (*Bower, Brown & Purves, 1978*), pea seedlings (*Miyata et al., 1981*; *Zdunek-Zastocka, 2010*), and maize (*Zea mays* L.) coleoptiles (*Koshiba et al., 1996*). Both animal and plant AOs possess relatively wide substrate specificity and can oxidize a number of different aldehydes, and this trait indicates that AOs have different biological roles (*Koshiba et al., 1996*; *Ori et al., 1997*; *Akaba et al., 1999*; *Omarov et al., 1999*; *Koiwai et al., 2000*, *2004*; *Seo et al., 2000a*; *Ibdah et al., 2009*). AO isoforms have been largely characterized in *Arabidopsis*, and the AO gene family consists of at least four aldehyde oxidase genes, AO1–AO4 by forming homodimers and heterodimers (*Akaba et al., 1999*; *Koiwai et al., 2000*, *2004*; *Seo et al., 2000a*; *Zdunek-Zastocka et al., 2004*).

In *Arabidopsis* AOα (homodimers of AAO1) showed a relatively high efficiency for indole-3-acetaldehyde, a precursor of IAA and it indicates AOα is a possible candidate aldehyde oxidase for IAA biosynthesis, and this activity is higher in IAA-overproducing *sur1* mutant seedlings (*Akaba et al., 1999*; *Seo et al., 1998*). The study of AO in maize

also confirms its role in IAA biosynthesis (*Koshiba et al., 1996*; *Sekimoto et al., 1997*). *Fedorova et al. (2005)* proposed that the local synthesis of IAA in the root nodule meristem and the modulation of AO expression and activity are involved in the regulation of *Lupinus albus* and *Medicago truncatula* nodule development. Two isoforms, BrAO-a, and BrAO-d, were suggested to be involved in auxin overproduction during clubroot development induced by pathogen infection in Chinese cabbage (*Ando et al., 2006*). In *Arabidopsis*, the expression patterns of AO1 are tissue specific and are expressed predominantly in seedlings, roots, and seeds.

Based on the publicly available microarray data (http://bar.utoronto.ca/eplant/, *Kilian et al., 2007*), we can predict AO1 might be involved in hypoxia response in *Arabidopsis* because it was induced significantly by hypoxia stress. Waterlogging or flooding is one of the main abiotic stresses that limit crops production and deficiency of oxygen is the main reason responsible for the damages caused by waterlogging or flooding stress, and according to the extent of lack of oxygen, it can be divided into anoxia (nearly no oxygen) or hypoxia (low oxygen). In most cases, waterlogging or flooding stress cause hypoxia conditions and it is detrimental to plants due to the energy crisis caused by anaerobic respiration and the accumulation of toxic substances (*Xu, Pan & Zhang, 2020*).

## Biochemical and physiological function of AO2

AO2 is highly expressed in hypocotyls, seedlings, roots and senescing leaves. In *Arabidopsis*, AO2 homodimers have a strong preference for 1-naphthaldehyde as a substrate, which is oxidized to 1-naphthyl acetic acid, whereas the AO1/AO2 heterodimer has a preference for indole-3-acetaldehyde and 1-naphthyl acetic acid (*Akaba et al., 1999*). The deduced amino acid sequence of aldehyde oxidase 2 (AhAO2) in *Arachis hypogaea* L. showed high similarity with other plant AOs. *AhAO2* was dominantly expressed in leaves, and its transcript level was greatly increased under exogenous ABA application; overexpression of *AhAO2* in *Arabidopsis* led to improved ABA levels and drought tolerance after drought treatment (*Yang et al., 2011*). ZmAO-2 in maize was expressed at a higher level in coleoptiles than in roots (*Sekimoto et al., 1997*).

From the prediction of the website, AtAO2 in *Arabidopsis* (AT3G43600) could interact with two proteins (gene number: AT1G19730 and AT5G42980) which are belong to thioredoxin superfamily and it indicates AO2 might involve in redox reaction.

## Biochemical and physiological function of AO3

Among the AOs, AO3 has gained great attention owing to its possible involvement in ABA biosynthesis and its importance under normal and stress conditions (*Seo et al., 2000a*, *2000b*; *González-Guzmán et al., 2004*). The AO3 gene can efficiently catalyse the final step in ABA synthesis in *Arabidopsis* leaves (*Seo et al., 2000b*) and seeds (*González-Guzmán et al., 2004*), in barley roots (*Omarov et al., 2003*), and in pea plant leaves and roots (*Zdunek-Zastocka et al., 2004*).

AO3 is mainly localized in vegetative organs such as the vascular tissues of roots, hypocotyls as well as leaves and is also slightly expressed within guard cells and surrounding cells (*Koiwai et al., 2004*; *Nambara & Marion-Poll, 2005*). Rice *OsAO3* was

expressed in germinated seeds, roots, leaves, and floral organs, particularly in vascular tissues and guard cells, and its expression was significantly induced by exogenous ABA and mannitol treatment. The *osao3* mutant accumulated less ABA and exhibited earlier seed germination, increased seedling growth and grain yield, and decreased drought tolerance than the wild-type, and *OsAO3*-overexpressing lines exhibited the opposite phenotype. Mutation and overexpression of *OsAO3* increased and decreased grain yield, respectively, by affecting panicle number per plant, spikelet number per panicle, and spikelet fertility (*Shi et al., 2021*).

In *Arabidopsis*, AO3 mRNA is mainly expressed and most AOd enzyme activity is detected in rosette leaves. The *ao3-1* mutant exhibited a wilted phenotype under normal conditions, reduced ABA levels accompanied by excessive water loss and retarded vegetative growth and reduced stress tolerance, but this mutant showed a less prominent dormancy-associated phenotype in seeds compared with *aba3-2*, with impaired function of all AOs due to the dysfunction of Moco (*Seo et al., 2000b*; *González-Guzmán et al., 2004*). These findings indicated that AOd is involved in ABA biosynthesis in leaves and that other aldehyde oxidases are involved in ABA biosynthesis in other organs, such as roots, silique, and seeds (*Seo et al., 2004*). Consistently, mutations in AO1 or AO4 in the *AO3* mutant background aggravated ABA deficiency in seeds, demonstrating that both gene products contribute partially to ABA biosynthesis and act as a 'back-up system' to AO3 in the AO3 mutant background, while this effect might be negligible in wild-type *Arabidopsis*. Absence of AAO3 oxidation activity (rather than the lower ABA) and its associated function is responsible for the earlier senescence symptoms in *aao3* mutant. *Arabidopsis* AAO3 knockout mutant *aao3* exhibited earlier senescence compared with wild-type during normal growth or upon application of UV-C irradiation. Different aldehydes accumulated prominently in aao3 mutants compared with WT leaves under normal growth conditions, upon UV-C irradiation and after exogenous aldehydes application (*Nurbekova et al., 2021*).

AO3 can be slightly induced by stresses; for example, an increase in abscisic aldehyde oxidizing activity was observed for the PsAO isoform in the leaves and roots of pea plants exposed to salinity or ammonium treatments (*Zdunek-Zastocka et al., 2004*). Drought stress increases AO3 transcript levels and the activity of AOd (*Seo et al., 2000b*; *Bittner, Oreb & Mendel, 2001*) and high AO3 expression was sufficient to induce stomatal closure (*Melhorn et al., 2008*).

## Biochemical and physiological function of AO4

A study on *Arabidopsis* AO4 knockout mutants and overexpressing lines demonstrated that AO4 plays a critical role in delaying senescence and protecting silique against toxic aldehydes (*Ibdah et al., 2009*). Single loss-of-function mutants for AO4 failed to show significant changes in endogenous ABA levels in seeds compared with wild type. Plants produce toxic aldehydes under normal and stress conditions (*Mano, 2012*; *Srivastava et al., 2017*) and excess aldehydes may lead to reactive oxygen species (ROS) accumulation that could damage proteins, lipids and other molecular compounds (*Bartels, 2001*; *Kotchoni et al., 2006*; *Zhang et al., 2012*; *Biswas & Mano, 2015*). AO4 was expressed most abundantly

**Table 1  Aldehyde oxidase in plants.**

| Species | Substrate/localization | Protein | Function | Reference |
|---|---|---|---|---|
| Cucumber | Benzaldehyde, Phenylacetaldehyde Indole-3-acetaldehyde (major) | AO1 | IAA biosynthesis | *Bower, Brown & Purves (1978)* |
| Pea | Indole-3-acetaldehyde | AO1 | IAA biosyntesis | *Miyata et al. (1981)* |
| Pea | Indole-3-aldehyde | AO3 | AO activity and ABA induced by suboptimal conditions | *Zdunek & Lips (2001)* |
| Maize | Indole-3-acetaldehyde/apical region of coleoptiles | AO1 | IAA production | *Koshiba et al. (1996)* |
| Barley | indole-3-aldehyde, acetaldehyde, heptaldehyde, benzaldehyde | AOs | | *Omarov et al. (1999)* |
| Oat | indoleacetaldehyde | | IAA biosynthesis | *Rajagopal (1971)* |
| Arabidopsis | Indole-3-aldehyde | AO1 | ABA biosyntesis | *Seo et al. (1998)*, *Böttcher et al. (2014)* |
| Arabidopsis | Abscisic aldehyde/rosette leaves | AO3 | ABA production | *González-Guzmán et al. (2004)* |
| Arabidopsis | Abscisic aldehyde | AO3 | ABA and ROS production, drought and water stress | *Yesbergenova et al. (2005)* |
| Arabidopsis | Aromatic and aliphatic aldehydes/silique | AO4 | Aldehyde detoxification | *Srivastava et al. (2017)* |
| Arabidopsis | Abscisic aldehyde, hexanal, and acetaldehyde | AO3 | ABA production, Aldehyde detoxification | *Nurbekova et al. (2021)* |
| Arabidopsis | Abscisic aldehyde/leaves | AO3 | ABA biosyntesis | *Seo et al. (2000a)* |
| Arabidopsis | Abscisic aldehyde/seeds | AO3 | ABA biosyntesis | *Seo et al. (2004)* |
| Arabidopsis | Benzaldehyde/sillique | AO4 | Benzoic acid | *Ibdah et al. (2009)* |
| wheat | Abscisic aldehyde | AO3 | carotenoid pigments accumulation | *Colasuonno et al. (2017)* |
| Rice | Abscisic aldehyde/germinated seeds, roots, leaves, floral organs, vascular tissues, guard cells | AO3 | ABA production, drought stress | *Shi et al. (2021)* |

in developing silique, and AO4 was involved in the oxidation of benzaldehyde in silique, contributing to the synthesis of the benzoic acid (BA) pool in seed capsules (*Ibdah et al., 2009*). The AO4 mutant seeds showed significant reductions in the total contents of benzaldehyde and 4-benzoyloxybutylglucosinolate, as well as a slight reductions in the levels of 3-benzoyloxypropylglucosinolate (*Ibdah et al., 2009*).

AO4 can oxidize various aromatic and aliphatic aldehydes and it differentially generates superoxide ($O^{2-}$) and hydrogen peroxide ($H_2O_2$) in an aldehyde-dependent manner. In addition, AO4 transcript levels and activity in silique were induced by $H_2O_2$ application, indicating that the ROS generated by the activity of AO4 can self-amplify, enabling further detoxification of toxic aldehydes and delaying silique senescence of *Arabidopsis* plants (*Srivastava et al., 2017*). Functions of aldehyde oxidases in plants were summarized in Table 1.

## SUMMARY AND PROSPECT

Phylogenic tree showed at least two conserved clades for AO enzymes involved in the biosynthesis of ABA and auxin synthesis (*Abu-Zaitoon, 2014*) and it supports the proposed function of ABA and auxin biosynthesis. Studies about AOs are quite scant; however, AOs are very import enzymes merit better understandings due to two reasons. Firstly, they

may involve in biosynthesis of important phytohormones, and it may be a potential target for genetic control in agriculture as genes like GA20-oxidase in green revolution. Phytohormones are vital substances regulates multifaced function of plants and mutants in the early steps of biosynthetic pathway could be lethal and useful tools in scientific researches while mutation in late nodes could be important regulating points and easier for genetic manipulation in agriculture. Secondly, AOs could have multifaced functions due to their wide substrate of AOs, thus pathway or traits influenced by AOs also deserve further studies. In this review, we summarized the recent studies about AO in plants including the motif information, biochemical, and physiological functions which may provide useful information for botanist and biologist and it is also beneficial for agriculture.

AO may be involved in metabolic processes not only phytohormone synthesis but also cellular ROS homeostasis under normal and stress conditions (*Wendehenne et al., 2014*; *Gilroy et al., 2014*). Recombinantly expressed *Arabidopsis* AO1 and AO3 in the methylotrophic yeast *Pichia pastoris* are capable of producing not only $H_2O_2$ but also $O^{2-}$ during oxidation of aldehydes (*Zarepour et al., 2012*). In addition to their aldehyde oxidation activity, AO1 and AO3 were found to exhibit NADH oxidase activity (*Yergaliyeva et al., 2016*; *Brikis et al., 2018*; *Zhong et al., 2010*; *Hesberg et al., 2004*; *Diallinas et al., 1997*; *Yesbergenova et al., 2005*). In this respect, conditions such as natural senescence and stress-induced stomatal movement, which both require simultaneously elevated levels of abscisic acid and hydrogen peroxide/superoxide, are likely to benefit from AOs by formation of abscisic acid and ROS.

As mentioned above, Moco is an important component of several enzymes including nitrate reductase, AOs, XDH. The activity of AO enzymes requires a sulfurated Moco, which is converted from the desulfo- to the sulfo-form by the Moco sulfurase ABA3 (*Xiong et al., 2001*; *Bittner, Oreb & Mendel, 2001*). Deficiency in the biosynthesis of Moco might result in pleiotropic effect or even lethal consequences for the respective organisms (*Mendel, 2013*). For example, Moco deficient mutants of barley *Az34* (*Walker, Kudrna & Warner, 1989*), tobacco (*Leydecker et al., 1995*), tomato *flacca* and *sitiens1* (*Sagi, Fluhr & Lips, 1999*; *Sagi, Scazzocchio & Fluhr, 2002*), and *Arabidopsis aba3* (*Akaba et al., 1999*; *Bittner, Oreb & Mendel, 2001*) had severely impaired ABA production.

The availability of Moco and its sulfuration may therefore represent a potential point of interaction between cytokinin, ABA and IAA metabolism in plants to control phytohormone balance, which may determine the stress tolerance of plants (*Cowan & Tayor, 2001*). In addition, the plant Moco-pool size (enzymes with Moco) varies under different nutritional and environmental conditions (*Sagi et al., 1997*; *Sagi, Omarov & Lips, 1998*). Thus, increased activity of Mo-hydroxylases (*i.e.*, AO and XDH) in response to salt stress and ammonium treatment could be one mechanism for stress adaptation, which could include enhanced activity of the Moco-containing enzymes responsible for ABA synthesis (*Sagi, Omarov & Lips, 1998*). When XDH activity or nitrate assimilation is reduced or inhibited, more Moco might therefore be available for sulfuration (*Sagi, Fluhr & Lips, 1999*; *Sagi, Scazzocchio & Fluhr, 2002*; *Bittner, Oreb & Mendel, 2001*) and thus AOs enzymes were more activated. One the other hand, when nitrate reductase has been

induced by its substrate, the AO activity and ABA content in plants were reduced (*Omarov, Sagi & Lips, 1998*). Molybdenum as one of the trace essential elements for plants, and the effect of external sources molybdenum on AO, XDH, and other enzymes as well as phytohormone biosynthesis could be a promising research direction.

In addition, AO1 was predict to involve in hypoxia stress of *Arabidopsis* and AO2 was assumed to interact with proteins belong to thioredoxin superfamily. We can speculate AOs might regulate the abiotic stress of plants including drought, and osmotic, stress. The study about AO enzymes thrived around 20 years ago, and they need further study for possible manipulation in agriculture for suitable phytohormone biosynthesis, ROS homeostasis, stress tolerance, development, and aldehyde detoxification.

# ACKNOWLEDGEMENTS

We thank Dr. Qisen Zhang for useful discussion.

## Funding

The work was supported by the funding from the National Natural Science Foundation of China (31901438) and crop varietal improvement and insect pests control by nuclear radiation program. The funders had no role in study design, data collection and analysis, decision to publish, or preparation of the manuscript.

## Grant Disclosures

The following grant information was disclosed by the authors:
National Natural Science Foundation of China: 31901438.

## Competing Interests

The authors declare that they have no competing interests.

## Author Contributions

- Jun Wu performed the experiments, authored or reviewed drafts of the paper, and approved the final draft.
- Blair Moses Kamanga analyzed the data, authored or reviewed drafts of the paper, and approved the final draft.
- Wenying Zhang analyzed the data, authored or reviewed drafts of the paper, and approved the final draft.
- Yanhao Xu conceived and designed the experiments, prepared figures and/or tables, authored or reviewed drafts of the paper, and approved the final draft.
- Le Xu conceived and designed the experiments, performed the experiments, prepared figures and/or tables, authored or reviewed drafts of the paper, and approved the final draft.

## Data Availability

There is no data to publish; this article is a literature review.

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
