# Peer review of "Research progress of aldehyde oxidases in plants"

_PeerJ, doi:10.7717/peerj.13119_

## Round 0.1 · original submission · Major Revisions

Please provide a comprehensively revised version addressing the editorial comments and a detailed rebuttal letter.

·

Basic reporting

The manuscript summarizes recent studies about AO in plants considering genes and enzymes involved in synthesis and function that is in the scope of the Journal. Although authors made a substantial revision of the bibliography, considering the number of recent investigations of AO in specific plants, authors need to integrate the information and include a discussion to establish further investigations or points of view. In addition, authors must use figures and diagrams to support the information in text and point out the missing information or opportunity areas for new research. All of these suggestions could help to increase the audience of the manuscript.

Experimental design

The authors declare the use of Web of Science (Clarivate Analytics). Keywords such as “aldehyde oxidase,” “abscisic acid,” “indoleacetic acid,” and “stress. Considering the broad range objective of the manuscript, other keywords should be included to search among plants, mechanism, regulation, inhibition, etc. that helps to integrate the information in the manuscript.
The sources are adequately cited, but the organization of the review needs to be corrected in some paragraphs. Example: 66-68 indicates that details in AO in IAA production will be discussed later, but is not clear in the manuscript. There is a lack of paragraph connection in lines 104-105. It should be clear when authors make their personal interpretation or discussion (Example line 125).

Validity of the findings

The introduction requires some structure corrections to better meet the goal of the manuscript. The authors dedicate an important part of the introduction to the zeaxanthin epoxidase enzyme function and precursors, which miss the idea covered in each subject of the manuscript. Also, the conclusion should include unresolved questions or further directions. The authors only dedicate three lines to the conclusion.

Additional comments

The abstract should be improved in the specific points of view and discussion of the review.
In line, 38 authors indicate that ABA genes are identified in various plants but only describe enzymes characteristics and not gene regulation.
I suggest the use of a figure or pathway diagram to better support the idea described in lines 40 to 46.
It is not clear if there is a lack of information regarding AOs or the existing information is not clear enough to support the two reasons described in the 50-54 lines.
Line 68 - It is not clear which pathway the authors made reference.
Line 75-77 avoid independent and short ideas.
Line 91-93- ¿Incomplete or missing idea?
Line 95-97 is not necessary to explain.
104 – 105 lack of paragraph connection – miss the main idea – substrate specificity, structure, and expression?
125-125 It is not clear if the sentence is an author idea or hypothesis
126-127 The website has many tools, and it is not clear how the authors made the hypoxia prediction.
Line 128 - In contrast with other AO described in the manuscript the information is scarce.
The table included does not support the manuscript. I suggest a figure or pathway diagram
The manuscript has a good structure, but I found some writing issues related to determiner use (a/an/the/this, etc.), comma misuse within clauses, passive voice misuse. I suggest a revision to improve clarity.

Reviewer 2 ·

Basic reporting

The authors need to pay attentions to the English language and use grammatically correct sentences, especially in lines 50-56 require both spelling and grammar checks.
The review article has referenced a significant number of articles studying the biosynthesis of ABA/IAA involving aldehyde oxidases. The basic properties of AO’s catalysis and physiological functions are clearly presented allowing general audience to comprehend before detailed discussion of each AO isoform in plants. The authors can strengthen their arguments about how this review article will impact the indicated fields in lines 50-56 of introduction.
A figure depicting the basic biochemical or physiological processes of AOs will help readers with different background appreciate this work better and easier. Additionally, a table summarizing the differences and similarities among AO isoforms will serve the same purpose.

Yes, the review has broad and cross-disciplinary interest and within the scope of the journal.
No, the field hasn’t been reviewed lately.
Yes, Introduction adequately introduce the subject. The motivation of this review could be strengthened as mentioned above.

Experimental design

Yes, the content is within the Aims and Scope of the journal.
Yes, the Survey Methodology is consistent with a comprehensive, unbiased coverage of the subject.
Yes, sources are adequately cited and appropriate.
Yes, the review is organized logically into coherent paragraphs/subsections

Validity of the findings

In the last session of the article extends beyond the well-characterized functions of AO and four AO isoforms, so the session shouldn’t be named as ‘Summary’. On the other hand, the benefits of this work stated in lines 50-56 are not only summarizing the current advances of the discussed field but also providing useful information to agriculture, whereas this session falls short of intended discussion and implications. Alternatively, a future direction of AO research may be discussed in the ‘summary’ session.

Reviewer 3 ·

Basic reporting

(1) In the basic information, the biosynthetic processes of IAA and ABA are explained. However, it is difficult to understand only the text description. Please illustrate the biosynthetic process including chemical structures and clearly show the pathways in which AO is involved.

(2) In addition, the biochemical and physiological functions of AO1 to AO4 are summarized, but it would be easier to understand if the localization, characteristic substrates, and physiological functions in plant are summarized in a table.

Experimental design

No comment

Validity of the findings

No comment

Additional comments

In the introduction, "GA20-oxidase" is explained, but please add more details about its potential as a target for gene regulation in agriculture.

In the biochemical and physiological function of AO1, it is stated that AO1 may be involved in the hypoxic response because it is induced by hypoxic stress. Please add more explanation about the physiological significance of hypoxic response in plants.

---

## Round 0.2 · accepted · Accept

Thanks for addressing the minor revisions requested. Now your manuscript is accepted in PeerJ.

·

Basic reporting

The manuscript summarizes recent studies about AO in plants considering genes and enzymes involved in synthesis and function that is in the journal's scope. The authors made substantial modifications to improve the manuscript structure, adding references and figures to integrate the information. Also made all the modifications suggested in the review document.

Experimental design

No comments

Validity of the findings

The authors added more information in the introduction section that makes it easier to understand the manuscript goal and unanswered questions.

Additional comments

The authors improved the abstract, including specific points of view and discussion of the review, and corrected it according to all the suggestions.

Reviewer 2 ·

Basic reporting

no comment

Experimental design

no comment

Validity of the findings

no comment

Reviewer 3 ·

Basic reporting

No comment

Experimental design

No comment

Validity of the findings

No comment

Additional comments

No comment